# Finding Common Ground:
# Annotating and Predicting Common Ground in Spoken Conversations

**Magdalena Markowska**[♣♢‡]**, Mohammad Taghizadeh**[⋆†]**, Adil Soubki**[⋆♢‡]**,**
**Seyed Abolghasem Mirroshandel**[⋆†]**, Owen Rambow**[♣♢‡]
[⋆] Department of Computer Science [♣] Department of Linguistics
[♢] Institute for Advanced Computational Science
[‡] Stony Brook University, Stony Brook, NY, USA
[†] University of Guilan, Rasht, Guilan, Iran
Corresponding Author: magdalena.markowska@stonybrook.edu

## Abstract

When we communicate with other humans, we do not simply generate a sequence of words. Rather, we use our cognitive state (beliefs, desires, intentions) and our model of the audience's cognitive state to create utterances that affect the audience's cognitive state in the intended manner. An important part of cognitive state is the common ground, which is the content the speaker believes, and the speaker believes the audience believes, and so on. While much attention has been paid to common ground in cognitive science, there has not been much work in natural language processing. In this paper, we introduce a new annotation and corpus to capture common ground. We then describe some initial experiments extracting propositions from dialog and tracking their status in the common ground from the perspective of each speaker.

## 1 Introduction

Expressions such as "finding common ground" or "establishing common ground" are familiar to everyone. They are often used with reference to two or more people trying to reach some sort of mutual understanding about various topics; for example, politicians during a parliamentary meeting. Most often, however, common ground does not involve a conscious effort. Establishing and updating common ground (CG) between interlocutors is the key to a successful conversation. It is therefore crucial to develop methods of CG representation and learning. This work focuses on modelling the concept of common ground from the cognitive perspective. Specifically, we introduce the first (to the best of our knowledge) corpus annotated for common ground. The procedure developed for the annotation is designed such that it reflects what we think happens when people update their common grounds. We further conduct preliminary machine learning experiments focused on extracting events subject to CG updates, and on predicting the CG

updates from the perspective of the two speakers in a dialog.

The concept of common ground has been widely studied across various disciplines including linguistics and cognitive science (Kiparsky and Kiparsky, 1968; Karttunen, 1971a,b; Clark, 1996; Horton and Gerrig, 2016), computer science, artificial intelligence, and natural language processing (Grosz and Sidner, 1990; Cohen and Levesque, 1990; Traum, 1994; Del Tredici et al., 2022), and philosophy (Lewis, 1969; Stalnaker, 2002). CG can be described in simple terms as a set of beliefs which interlocutors believe they share, and which they believe other discourse participants also believe they share. However, when people communicate, they do not only exchange their beliefs, but also other components of cognitive state, such as desires, intentions, goals, etc. Consequently, the full definition of CG should also contain sets of shared desires, intentions, and so on. Because this is the first corpus created for that purpose, we focus solely on the belief component of CG. We do acknowledge, however, that future work will need to extend the notion of CG.

This paper is organized as follows. We summarize the NLP literature in Section 2. We present our new corpus in Section 3. In Sections 4, 5, and 6 we discuss our baseline experiments for predicting the events, beliefs about events, and achieving common ground.

## 2 Related Work

The concepts of belief and factivity are closely related, see (Prabhakaran et al., 2015, Section 2) for a discussion. They have been annotated in multiple language corpora, such as LU (Diab et al., 2009), FactBank (Saurí and Pustejovsky, 2009), UW (Lee et al., 2015), LDC CB (Prabhakaran et al., 2015), MegaVeridicality (White et al., 2018), CommitmentBank (de Marneffe et al., 2019), RP (Ross and Pavlick, 2019), and BeSt (Tracey et al., 2022),

among others. While all of those datasets differ in terms of identification of events and the annotation procedure, they all target one common goal. Given some text, they identify the level of commitment to the events expressed in the text, whether that be from the perspective of the author solely (e.g. LU, MegaVeridicality, RP, UW) or from the author along with other mentioned sources (FactBank, BeSt).

What is considered an event is usually tailored to the specific question that a corpus is addressing. For example, CB targets solely final clausal complements as it examines the role of the matrix predicate in determining the speaker's belief, while LU and LDC CB consider all full lexical verbs in a sentence, and FactBank and BeSt target verbs and nouns referring to events and states. The genre of the data is also correlated with the task. FactBack uses WSJ newswire, and as a result the author is limited in the expressiveness of their cognitive state, as news-writing requires following certain standards. RP, LU, and CB, on the other hand, use mixed genres, including discussion forums and blogs, which allow for more creative and expressive freedom. All of these corpora also exhibit various scales identifying the level of belief. Those are either represented numerically, averaging over annotators' judgements, on a [-3,3] scale (UW, CB, RP), or with categorical labels (LU, FactBank, LDC CB, BeSt). One last crucial factor affecting the outcome of annotation are the annotators themselves. RP and UW use crowdsourced annotators, while FactBank, LU, LCD CB, and BeSt opt for trained annotators.

Belief/factivity prediction tasks have attracted a lot of attention in the past two decades. In the early 2000s, rule-based systems were used to detect authors' beliefs (Nairn et al., 2006). Diab et al. (2009), Prabhakaran et al. (2010), and Lee et al. (2015) use SVMs together with dependency trees and lexical based features. Newer, neural network approaches include bidirectional LSTMs for both single and multi-task setups (Rudinger et al., 2018). Pouran Ben Veyseh et al. (2019) use BERT representations with graph convolutional neural networks. Jiang and de Marneffe (2021) use fine-tuned BERT for belief detection on various corpora and closely examine shortcomings of the model. Murzaku et al. (2022) show that combination of specific corpora can improve the model's performance.

CG is closely related to Theory of Mind (ToM),

a concept first discussed in Premack and Woodruff (1978), which refers to the capacity in humans (Valle et al., 2015), as well as chimpanzees and orangutans (Call and Tomasello, 1998), to infer information about others' mental states including beliefs, desires, and emotions. CG is an aspect of ToM, since beliefs in the CG are beliefs a speaker models as being held by the addressee. This means that humans (as well as other primates) are able to infer some "hidden" information about what other people think, believe, and perceive in particular situations. A natural question for NLP is whether contemporary advanced language models exhibit knowledge of ToM. Furthermore, if not, what are some possible approaches that will allow machines to infer information about their conversational partner's cognitive states?

Kosinski (2023) tested current large language models (LLMs) by having them complete narratives inspired by the Sally-Anne (Wimmer and Perner, 1983; Baron-Cohen et al., 1985) and Smarties (Perner et al., 1987) tests, both of which were originally designed to determine if children are aware of the cognitive state of others. He focused specifically on one subcomponent of the ToM scale (Wellman and Liu, 2004), namely false belief. The results found that the most advanced GPT models correctly complete both true and false beliefs for characters in various scenarios. For example, if there is a bag of popcorn with a "chocolate" label on it, the models accurately predict that when Sam finds the bag she will believe that the bag is full of chocolate. Kosinksi then concludes that this is either (1) evidence of the inadequacy of these tests for evaluating ToM or (2) evidence of ToM emerging as a byproduct of learning on different tasks. Ullman (2023) disagrees with this hypothesis and states that it is indeed possible that ToM tests are in fact accurate and useful for studying human cognition but not LLMs. His claim is backed by a series of ToM experiments, slightly different from the ones in Kosinski (2023), that show that LLMs fail to make correct false-belief predictions. For example, in the popcorn-chocolate bag scenario they added additional information stating that Sam cannot read. GPT 3.5 still guessed with 98% confidence that Sam believes the bag is full of chocolate. Similarly, Sileo and Lernould (2023) develop a dataset of "mind games" using dynamic epistemic logic and target aspects of ToM to show that contemporary LLMs fail to make correct inferences in

other scenarios as well.

The experiments conducted by Kosinski (2023), Sileo and Lernould (2023), and Ullman (2023) show that while LLMs *can* produce accurate completions for false-belief in *some* settings, the results are highly sensitive to perturbation and far from robust. This on its own should be sufficient evidence that ToM cannot possibly be an emergent property of LLM learning, at least not yet, and points to a need for higher quality, naturalistic data on how humans reason about aspects of ToM, like CG.

## 3 The CG Corpus

The Common Ground Corpus is annotated on the top of the LDC CALLHOME American Speech corpus (Canavan et al., 1997), which consists of collections of 120 unscripted dialogs between close friends or family members. The dialogs are available in both written form and audio. Since our dataset is the first attempt at annotating CG in a discourse, we chose to start with conversations between two people. Another crucial aspect of the CALLHOME corpus is that the interlocutors know each other very well, which allows us to assume that their CG set is non-empty at the start of the conversation. Finally, the speakers were allowed to speak freely about any topic, which allows us to capture the speakers' cognitive states in natural settings.

In a dialog, each of the interlocutors form their own cognitive states, which includes a model of other speakers' cognitive state. Since CG forms a substantial component of a speaker's cognitive state, we model CG for each speaker separately: the CG is not shared. In the majority of cases, the CGs of the two interlocutors converge. However, we also see scenarios in which the CGs for speakers diverge and in which we observe miscommunications.

The two main parts of the annotation procedure are event extraction, and belief and CG annotation.

### 3.1 Event extraction

CALLHOME dialogs are divided into utterances. The events are formed from the main predicates in each clause forming the utterance. To make the events as informative as possible, we resolve any pronominal anaphors, i.e. we spell out any referents of pronouns, including first and second person (sometimes resulting in intentionally unnatural sentences). Here is an example from the corpus.

**Example 1:** A: *I thought I was going to get to see everybody.*
$e_1$: A thought A was going to get to see everybody.
$e_2$: A was going to get to see everybody.
$e_3$: A got to see everybody.

There are two cases where we create events that are not explicitly mentioned in a speaker's utterance. First, we form speech act events to signal question asking or giving orders. Second, if one speaker elliptically negates an event, or gives a negative answer to an event under question, we create a fully fledged event out of it. Both cases are illustrated in the first three columns of Table 1.

### 3.2 Belief and CG annotation

As was discussed in Section 1, we limit the definition of CG to beliefs in this paper. In order to infer the interlocutors CG, we first identify their beliefs towards the formulated events. We follow the belief types proposed in FactBank (Saurí and Pustejovsky, 2009). Since the main goal of our corpus is to represent the CG updates, as opposed to the granularity of speakers' beliefs, we simplify the types and limit them to 4 different levels of belief. An event $e$ will be marked: **CT+:** if a speaker certainly believes that $e$; **CT-:** if a speaker certainly believes that not $e$; **PS:** if a speaker possibly believes that $e$; **NB:** if a speaker expresses no belief about $e$.

Once belief values are established for speakers A and B, the annotators determine the CG for each speaker. We differentiate between three distinct updates:

**JA:** an event $e$ is mutually believed by both interlocutors and is added to CG in the moment $e$ was uttered. The level of belief in the CG (which may differ from individual beliefs) is left implicit, since it is always the less certain degree of the two interlocutor's beliefs.

**IN:** an event $e$ has already been a part of the interlocutors' CGs before the time of the event. In other words, it must be true that at some point in the past, CG(A)=JA and CG(B)=JA for $e$.

**RT:** an event $e$ that has been presented by a speaker has been entertained but rejected by the addressee.

In order to reflect the dynamic nature of a dialog, we must allow changes to the interlocutors' beliefs and, consequently, their CGs. Our annotation interface allows us to conduct such updates and register the history. Table 1 shows how beliefs can change.

Our annotation procedure requires *looking*

| Nb | Utterance | e id | Event | Bel(A) | Bel(B) | CG(A) | CG(B) |
|---|---|---|---|---|---|---|---|
| 1 | A: So you've been leading the life of Reilly huh? | $e_1$ | A asks B if B has been leading the life of Reilly | CT+ $e_1$ | CT+ $e_1$ | JA $e_1$ | JA $e_1$ |
|  |  | $e_2$ | B has been leading the life of Reilly | PS $e_2$ | CT- $e_2$ |  |  |
| 2 | B: No. Not really. | $e_3$ | B has not been leading the life of Reilly | CT- $e_2$
CT+ $e_3$ | CT+ $e_3$ | RT $e_2$
JA $e_3$ | RT $e_2$
JA $e_3$ |

Table 1: An annotation sample.

*ahead*. An annotator, as an *overhearer* of the conversation, needs to have access to the interlocutor's responses to be able to determine the interlocutor's cognitive states. This is most evident in the case of questions. In the example in 1, in order to determine whether B believes $e_2$, the annotator must look to the next utterance to see that B's attitude towards $e_2$ is in fact **CT-**, already at the time of the question utterance (though A is not yet aware of B's belief towards $e_2$).

The event that is being questioned is $e_2$. Belief of a speaker asking about $e_2$ is determined by the structure of the question. In the example, A chooses to inquire about $e_2$ by using affirmative sentence structure and rising intonation. We identify it as expressing possible belief towards $e_2$. To determine the addressee's (B's) belief status, an annotator needs to continue reading the conversation until they see B's response in the second utterance. In Example 2, B negates $e_2$. This is marked as Bel(B)=CT- at the level of the event $e_2$ because B always believed certainly not $e_2$. Speaker A, on the other hand, needs to wait for B's response to potentially update their belief about $e_2$. A's belief about $e_2$ (Bel(A)=CT-) is recorded at the time of the second utterance. Now that both interlocutors' beliefs about $e_2$ are established, an annotator can deduce that this event will be rejected from CG from the perspectives of both A and B.

An obvious question to ask would be why one cannot determine the status of B's CG at the time of utterance 1 since both Bel(A) and Bel(B) are available. This is because establishing common ground is an iterative process, and it is not sufficient for B to have access to B's beliefs and A's beliefs about an event. B also needs to believe that A believes that B believes $e_2$ and the other way around, at a minimum. At the stage of $e_2$, B is in fact certain that A does not know that and needs to wait for B's response.

Finally, with B's negative response to A's question, we do not only update the speaker belief about $e_2$, but we also create a negated version of $e_2$ that will then enter CG in both interlocutors' mind unless it is further explicitly denied.

## 3.3 Inter-Annotator Agreement

Computing agreement for the event extraction task is challenging, because we need to penalize an annotation if the number of annotated events differs between annotators. We call this method EMBERT. Given a similarity measure $f(\cdot)$ which takes two events (strings) and returns a score $s \in [0, 1]$, we compute an event-matched score between the set of events extracted by an annotator taken to be the reference $E_r$ and the set of events extracted by an annotator for comparison $E_c$. We compute, among the possible mappings between $E_r$ and $E_c$, the mapping which maximizes the mean pairwise similarity as produced by $f(\cdot)$. In the event that $|E_r| \neq |E_c|$, any unmapped event from the annotator being compared receives a score of zero. Similarly, the annotator being compared receives a zero for each reference event which cannot be mapped. This penalty imparts high emphasis on annotators first finding the same number of events from each utterance and then producing events which have similar semantic meaning. The mean similarity for the maximal mapping is returned to produce an event-matched score. EMBERT uses cosine similarity between SBERT (Reimers and Gurevych, 2019) embeddings as the similarity measure $f(\cdot)$.

We report EMBERT among the four annotators for event extraction in Table 2. Scores for EMBERT exceed 0.7 among all pairs of annotators, with the exception of Annotator 2. We use these studies to decide which annotators to retain.

We evaluate inter-annotator agreement among the four annotators on the belief and CG annotations for both speakers using a pairwise Cohen's kappa (Cohen, 1960). We use gold events for this study. The mean of these results across the four tasks are reported in Table 3. According to Cohen, anything above 0.6 indicates substantial agreement, while values above 0.8 are considered "almost perfect agreement". We also computed computed Fleiss' kappa, which was 0.70 across all tasks (Fleiss, 1971). Based on the interpretation guidelines provided in Landis and Koch (1977) these numbers also indicate substantial agreement among the four annotators.

|          | Anno. 2 | Anno. 3 | Anno. 4 |
|----------|---------|---------|---------|
| **Anno. 1** | 0.56 | 0.73 | 0.79 |
| **Anno. 2** | - | 0.60 | 0.61 |
| **Anno. 3** | - | - | 0.84 |

Table 2: EMBERT inter-annotator agreement scores for event extraction.

|          | Anno. 2 | Anno. 3 | Anno. 4 |
|----------|---------|---------|---------|
| **Anno. 1** | 0.58 | 0.76 | 0.77 |
| **Anno. 2** | - | 0.63 | 0.60 |
| **Anno. 3** | - | - | 0.87 |

Table 3: Mean pairwise Cohen's kappa for Belief and CG judgments.

| Annotation Type |     | Train | Test |
|-----------------|-----|-------|------|
| Utterance       |     | 415   | 146  |
| Event           |     | 970   | 325  |
| Bel(A) + Bel(B) | CT+ | 1,576 | 523  |
|                 | CT- | 107   | 71   |
|                 | PS  | 150   | 34   |
|                 | NB  | 81    | 8    |
|                 | 0   | 26    | 14   |
| CG(A) + CG(B)   | JA  | 1,540 | 490  |
|                 | IN  | 124   | 58   |
|                 | RT  | 115   | 73   |
|                 | 0   | 161   | 29   |

Table 4: The distribution of different annotation types in the annotated dataset.

## 4 Experiments: Events

For generating events, we use the base version of FLAN-T5, which is an instruction-finetuned version of the T5 LLM (Chung et al., 2022). It has demonstrated impressive few-shot performance across various tasks, outperforming even larger models. In our experiments, the model receives utterances as input and it generates the corresponding event(s) for each utterance. We have fine-tuned FLAN-T5 on the training set, which is a small dataset (see Table 4), and evaluated the model on the test set.

Alongside the input utterance, the model can also receive contextual information as input. In this paper, the contextual information is provided as the following modes: 1) **Fixed window:** A pre-determined number of utterances preceding and/or following the target utterances or events will be included as input for the model. And 2) **Speaker-based window**: The model will receive all preceding and/or following utterances until it encounters an utterance from a speaker different from the speaker in the target utterance or event. The input format of the model is as follows:

```
"Preceding Context": {Preceding Utterances}
"Events": {Target Utterance}
"Following Context": {Following Utterances}
```

### 4.1 Experimental Setup

This version of the dataset consists of four dialogs. To assess the performance of our model, we selected three dialogs as the training set, while the remaining dialog was designated as the test set. The distribution of all annotation types available in the training and test partitions are shown in Table 4.

The experimental results for event generation are presented in Table 5, using the EMBERT measure introduced in Section 3.3.

| Models | EMBERT |
|--------|--------|
| FLAN-T5 No Context Event Generation | 45.94 |
| FLAN-T5 Speaker-Based Window (preceding) | 48.65 |
| FLAN-T5 Fixed Window Size 2 (preceding) | 48.51 |
| FLAN-T5 Fixed Window Size 4 (preceding) | **48.69** |

Table 5: Experimental results of Event Generation.

As it is shown in Table 5, we have examined FLAN-T5 with four different input formats: 1) No context: the model only receives the target utterance and generates its event(s), 2) Speaker-based window: in addition to the target utterance, the model also receives speaker-based utterances, 3) Fixed window size 2 (preceding): in addition to the target utterance, the model also receives the 2 preceding utterances, and 4) Fixed window size 4 (preceding): in addition to the target utterance, the model also receives the 4 preceding utterances. As can be seen, the model using four preceding utterances as context performed best by both metrics. We have explored alternative combinations, such as using the following context. However, these combinations yielded unsatisfactory results, which is why they are not reported.

### 4.2 Error analysis

We performed a preliminary error analysis on the first 60 utterances from the test sets in three conditions: without context, and with the context window of size two and four. We identified seven types of errors. **Senseless** errors are those which make no reference to the actual utterance. For instance, for the gold event *Jill's situation sounds like a real mess*, the system generated *The baby's birth mom's*

*mom mom's mom [ . . . ].* If an event was matching some part of what was said but still failed to provide sufficient information, it was marked **Intelligible**. **Anaphora** resolution error combines unresolved pronoun references and wrongly identified references, e.g. proper names in the test sets were mistaken with the ones from training sets. **Uninformative** errors were the results of arbitrary division of utterances per speaker or elided information in speakers' turns, e.g. *B says same car* instead of *B has the same car*. The remaining types are **Missing** events, i.e. not generated by the system, events wrongly generated by the system (**Gen**), and **Gold** errors.

Table 6 presents counts of each error type and the total number of errors given a particular context size. As expected, the more context our system receives, the better it performs. The number of missing events does not improve with context. This is somewhat expected given that most missed events were caused by some failure of pragmatic inference. Anaphora resolution errors are significantly reduced given more context, which was also expected behavior.

## 5 Experiments: Beliefs

For belief classification, we designed a comprehensive set of models utilizing BERT-base-cased (Devlin et al., 2019) and FLAN-T5. The system is trained to annotate beliefs for each speaker (e.g. Bel(A)=CT+) given a gold target event and any additional context. As with event extraction, we experiment with both fixed and speaker-based windowing methods and additionally investigate including forward context. In these experiments, utterances are presented to the model either as raw sentences or as the corresponding events of those utterances.

We fine-tune BERT and FLAN-T5 on the training set using the following input format:

```
"Preceding Context": {Preceding Events or
↪ Utterances}
"Target Event": {Target Event}
"Following Context": {Following Events or
↪ Utterances}
```

### 5.1 Data Augmentation

To mitigate the issue of data imbalance, particularly for the CT-, PS, and NB classes, we use a data augmentation technique. Given FLAN-T5's ability to handle multiple languages, we opt for a machine translation-based data augmentation approach. For the training set, we employ the FLAN-T5 model to translate events associated with the minority classes. These newly translated events are subsequently incorporated into the training set, followed by additional model fine-tuning. We utilize French, German, and Spanish translations of the events in this process.

### 5.2 Experimental Setup

The experimental results for belief classification are shown in Table 7. Measures of classification performance are computed for both Bel(A) and Bel(B) and this table reports the average. The utilized metrics include F1 for each possible label and overall accuracy. As the label distribution is highly imbalanced, we also report macro F1.

In Table 7, we have evaluated systems using BERT, FLAN-T5, and augmented FLAN-T5 with different contextual information. Except for one model (i.e., "FLAN-T5 Fixed Window 4 (2 preceding + 2 following)"), all models utilize preceding fixed or speaker-based windows with different sizes and the events of corresponding utterances are fed to the model as context. In cases where the context is provided in the form of "Utterance Context", the model receives the context as raw utterances without the inclusion of events.

The results reported in Table 7 highlight several important findings. (1) The problem at hand proves to be challenging due to the small number of examples for the minority classes, resulting in low macro F1 values. (2) In terms of macro F1, the FLAN-T5 models generally outperform the BERT-based models. In terms of accuracy, however, the best result is achieved by the speaker-based window using BERT. This is because BERT does particularly well with the very high frequency category **CT+**, and less well with the other categories. (3) Interestingly, the table reveals that the best results of fixed

| Input | Senseless | Intelligible | Missing | Anaphora | Uninformative | Gold | Gen: | Total |
|---|---|---|---|---|---|---|---|---|
| No context | 15 | 6 | 34 | 40 | 12 | 5 | – | 112 |
| Context 2 | 12 | 7 | 40 | 32 | 11 | 5 | 1 | 108 |
| Context 4 | 11 | 2 | 36 | 27 | 10 | 5 | 1 | 92 |

Table 6: Error analysis; counts per error category shown by context size of system

| Models | Bel(A,B) AVG | | | | | |
|---|---|---|---|---|---|---|
| | CT+ F1 | CT- F1 | PS F1 | NB F1 | Macro F1 | Accuracy |
| BERT1 Fixed Window Size 1 | 89.67 | 14.83 | 32.33 | 12.67 | 37.38 | 79.17 |
| BERT2 Fixed Window Size 2 | 90.00 | 17.33 | 29.83 | 11.50 | 37.17 | 79.50 |
| BERT3 Fixed Window Size 3 | 89.50 | 16.00 | 22.50 | 0.00 | 32.00 | 79.00 |
| BERT3 Speaker-based Window | **91.50** | 23.33 | 19.50 | 9.17 | 35.88 | **81.83** |
| FLAN-T5 Fixed Window Size 2 (Utterance Context) | 87.00 | 20.25 | 20.25 | 10.50 | 34.50 | 76.75 |
| FLAN-T5 Speaker-based Window (Utterance Context) | 86.17 | 11.25 | 18.67 | 5.50 | 30.40 | 73.75 |
| FLAN-T5 Fixed Window Size 1 | 87.00 | **35.00** | 28.50 | 13.33 | 40.96 | 74.67 |
| FLAN-T5 Fixed Window Size 2 | 86.83 | 34.67 | 28.67 | 17.50 | 41.92 | 74.67 |
| FLAN-T5 Fixed Window Size 3 | 85.50 | 27.00 | 26.00 | **20.00** | 39.63 | 72.00 |
| FLAN-T5 Fixed Window Size 4 | 87.50 | 27.50 | 25.50 | 11.00 | 37.88 | 74.00 |
| FLAN-T5 Fixed Window 4 (2 preceding + 2 following) | 87.75 | 27.50 | 30.75 | 12.75 | 39.69 | 76.25 |
| FLAN-T5 Speaker-based Window | 87.67 | 32.67 | 30.67 | 17.83 | 42.21 | 76.33 |
| Augmented | | | | | | |
| FLAN-T5 Fixed Window Size 2 (French) | 87.50 | 26.50 | 26.50 | 16.17 | 39.17 | 75.00 |
| FLAN-T5 Fixed Window Size 2 (French-German) | 88.33 | 34.00 | **32.67** | 19.00 | **43.50** | 77.83 |
| FLAN-T5 Fixed Window Size 2 (French-German-Spanish) | 88.50 | 32.00 | 31.75 | 8.25 | 40.13 | 77.00 |
| FLAN-T5 Speaker-based Window (French) | 87.25 | 24.25 | 28.50 | 19.25 | 39.81 | 74.25 |
| FLAN-T5 Speaker-based Window (French-German) | 88.50 | 15.67 | 26.50 | 8.50 | 34.79 | 75.83 |
| FLAN-T5 Speaker-based Window (French-German-Spanish) | 88.67 | 32.33 | 28.67 | 11.83 | 40.38 | 76.00 |

Table 7: Experimental results of Belief Classification

window based contexts are achieved when utilizing a window size of 2. Moreover, increasing the window size negatively impacts the macro F1 scores. Also, using the following context does not help the model. (4) Generally, the speaker-based window contexts exhibit superior performance compared to the fixed window approach and the utterance context approach. (5) The results also demonstrate the effectiveness of data augmentation, particularly when using French and German for augmentation. The augmented data contributes to achieving the best macro-averaged F1 results because of the minority classes.

## 5.3 Error analysis

We conducted error analysis of FLAN-T5 in the Speaker-based window condition. There were 92 errors in belief prediction, given the gold events. We differentiated between five types of errors. The most common errors were observed in events embedded under matrix predicates, such as *say*, *tell*, *hope*, *think*, or conjunctions such as *before* or *because*. For example, an event *A's sons never treat one another like A and B's mom and dad treat A* embedded under *hope* (A hopes that $e$), had predicted CT+ values for both speakers, while it should have been marked as only possible belief (PS). We call those type of errors **key words (KW)**. Another type of event that was problematic were events embedded in questions (**Question**). For example, the event *B sleeps* which is a part of a speech act *A asks B when B sleeps* has predicted values PS for

both speakers, when in fact this event is a presupposition that has already been part of the speakers' common ground, and should therefore be annotated as CT+. Jokes, sarcasm and hypotheticals form another error type that we named **hard inferences (HI)**. There were also cases of wrong predictions that could not be linguistically motivated. Those form fourth group called **other**. Finally, there were also a few **gold** errors that were the result of data formatting. Table 8 presents counts of errors within each type.

| KW | Question | HI | Other | Gold |
|---|---|---|---|---|
| 38 | 21 | 12 | 11 | 10 |

Table 8: Error types in the belief prediction task

## 6 Experiments: Common Ground

We implemented two strategies to predict common ground. The first strategy is based on heuristics and the second strategy is based on fine-tuning of the FLAN-T5 model.

**Heuristic based approach** In this strategy, we have utilized the following rules. We always update CG for both speakers with these simple heuristics.

1) If **Bel(A) = CT-** or **Bel(B) = CT-**, then **CG = RT**.
2) If **Bel(A) = CT+** and **Bel(B) = CT+**, then **CG = JA** or **CG = IN**.
3) If **Bel(A) = PS** and **Bel(B) = CT+**, then **CG = JA(PS)** or **CG = IN**.

| Models | CG(A,B) AVG | | | | |
|---|---|---|---|---|---|
| | JA F1 | IN F1 | RT F1 | Macro F1 | Accuracy |
| FLAN-T5 (No Event, No Context) | 94.50 | 0.00 | 99.50 | 64.67 | 90.50 |
| FLAN-T5 Fixed Window 0 (No Context) | 94.50 | 33.75 | 99.50 | 75.92 | 91.00 |
| FLAN-T5 Fixed Window 2 | 95.00 | 50.17 | 99.50 | 81.56 | 91.83 |
| FLAN-T5 Fixed Window 4 | 94.33 | 48.83 | 99.50 | 80.89 | 90.67 |
| FLAN-T5 Fixed Window 8 | **95.00** | **54.00** | **99.50** | **82.83** | **91.83** |
| FLAN-T5 Fixed Window 4 (2previous-2next) | 92.50 | 41.00 | 99.50 | 77.67 | 88.50 |

Table 9: Experimental results of FLAN-T5 based Common Ground classification utilizing Gold belief.

| Models | CG(A,B) AVG | | | | |
|---|---|---|---|---|---|
| | JA F1 | IN F1 | RT F1 | Macro F1 | Accuracy |
| FLAN-T5 Fixed Window 0 (No Context) | 88.00 | 28.25 | 0.00 | 38.75 | 78.00 |
| FLAN-T5 Fixed Window 2 | **88.67** | **55.83** | 15.83 | **53.44** | **79.17** |
| FLAN-T5 Fixed Window 4 | 87.00 | 42.75 | 23.00 | 50.92 | 76.25 |
| FLAN-T5 Fixed Window 8 | 87.00 | 46.00 | 21.50 | 51.50 | 76.50 |
| FLAN-T5 Fixed Window 4 (2previous-2next) | 87.00 | 38.50 | **27.25** | 50.92 | 76.50 |

Table 10: Experimental results of FLAN-T5 based Common Ground classification without utilizing Gold belief.

4) If **Bel(A) = CT+** and **Bel(B) = PS**,
then **CG = JA(PS)** or **CG = IN**.

5) If **Bel(A) = NB** or **Bel(B) = NB**, then **CG = NULL**.

Rules 2-4 under-determine whether the belief is already in the CG or newly added. In this context, the crucial task is to determine whether the target event had already been present in the common ground of the speakers (i.e., **CG = IN**) or not (i.e, **CG = JA**).

To address this problem, we first create a data structure called "dialog memory", which combines each event, its related (gold) beliefs, and extracted common grounds. When processing the "target event", all preceding events, beliefs, and common grounds are presented in the dialog memory. Then, within the dialog memory, the heuristic algorithm searches for the "target event" based on the SBERT similarity measure. For similarities more than a threshold, the "target event" will be regarded as an event that is already in common ground. Note that in this approach, the heuristic utilizes gold annotated beliefs.

**Common Ground classification with FLAN-T5 model** In the next approach, we used the FLAN-T5 model to classify common ground. We fine-tuned FLAN-T5 on the training set. The model takes the following input prompt format:

```
["Bel(A)": {Gold Bel(A)} "Bel(B)": {Gold
↪ Bel(B)}]
"Input Event with Context:"
"Preceding Context": {Preceding events}
"Target Event": {Target Event}
"Following Context": {Following events}
```

In this approach, it is important to note that the

| SBERT Threshold | CG(A,B) AVG | | | | |
|---|---|---|---|---|---|
| | JA F1 | IN F1 | RT F1 | Macro F1 | Accuracy |
| 0, 0.2 | 69.52 | 20.00 | 98.59 | 62.70 | 60.84 |
| 0.4 | 70.00 | 19.05 | 98.59 | 62.55 | 61.17 |
| 0.6 | 73.48 | 20.59 | 98.59 | 64.22 | 64.72 |
| 0.8 | 85.03 | **20.93** | 98.59 | 68.18 | 77.67 |
| 0.9 | 92.03 | 13.33 | 98.59 | 67.98 | 87.06 |
| 0.92 | 93.10 | 15.00 | **98.59** | **68.90** | 88.67 |
| 0.95 | 93.39 | 0.00 | 98.59 | 63.99 | 89.00 |
| 1 | **94.41** | 0.00 | 98.59 | 64.33 | **90.61** |

Table 11: Experimental results of heuristic based Common Ground classification utilizing Gold belief.

FLAN-T5 model operates in two distinct modes. The first mode incorporates gold annotated beliefs, while the second mode solely focuses on the events without utilizing belief information. In either approach it uses gold events.

### 6.1 Experimental Setup

The experimental results for CG classification using heuristics approach, based on different SBERT thresholds, are shown in Table 11. The experimental results for CG classification using FLAN-T5 with and without gold belief information are shown in Tables 9 and 10, respectively. In these tables, different contextual information has been studied. Furthermore, as demonstrated in Table 9, we conducted experiments on CG classification by exclusively utilizing gold belief information, excluding event and context information (i.e., "No Event, No Context" case). Additionally, we explored another scenario where the model operates without access to context information (i.e., "Fixed Window 0 (No Context)" case).

The results presented in Table 11 indicate the

inherent difficulty in designing a heuristic even with the inclusion of gold beliefs, as they show relatively low performance across the evaluated metrics. Comparing the performance of the FLAN-T5 model with gold beliefs and the heuristic, it is evident that the FLAN-T5 model outperforms the heuristic in all metrics. This highlights the superiority of the FLAN-T5 model when leveraging gold beliefs for CG classification.

Interestingly, when considering the FLAN-T5 model with gold beliefs, increasing the context window size yields better results. However, this trend is not observed in the case of the FLAN-T5 model without gold beliefs. Furthermore, it is worth noting that following utterances do not have a positive impact on context generation classification in both versions of the FLAN-T5 model.

As anticipated, the task of CG classification without gold beliefs proves challenging due to class imbalance.

## 6.2 Error analysis

Error analysis for CG prediction was conducted on FLAN-T5 with window size of 8. Out of 29 mistakes (311 events in total), 27 were of one kind, namely IN was mistaken for JA and the other way around (13:14 ratio). Such errors are expected as the IN condition requires either keeping track of events that have entered CG during the span of the conversation, or recognizing certain lexical items that indicate that those event have been already part of CG. The remaining two errors involved cases of one speaker mishearing the other and CG updates that were ignored for the moment.

## 7   Future work

There are many obvious extensions of our work to date. We list a few.

**Audio Modality** In our modeling, we only use the CALLHOME transcripts. However, intonation in English is related to what the speaker believes the hearers already know. We will incorporate the audio signal into future CG predictions,and will explore multimodal neural architectures.

**Existing Belief Corpora and Systems** We intend to incorporate existing belief corpora and systems into the belief prediction task, which could improve the system performance. Specifically, we intend to use the state-of-the-art FactBank-only belief system from Murzaku et al. (2023) as a belief tagger, and give those outputs to our system. We also intend to experiment with a multi-task learning (MTL) setup, where we jointly model belief and common ground updates.

**Graph Approaches** Our data can naturally be represented as a graph structure where speaker A and B's utterances are nodes, and the changes in belief and common ground being edge labels. We intend to leverage graph neural network approaches inspired by neural architecture and general insights from Zhang et al. (2023), who model the emotion recognition in conversation (ERC) task.

**Higher-Order Beliefs** We explicitly model first-order beliefs and the CG. Many higher-order beliefs follow from the combination of individual belief and CG – for example, if A says that Mary is coming to dinner but B rejects it, it does not enter A's model of the CG. As a result, we can infer a higher-order belief, namely that A believes B does not believe the event (since if B believed it, it would be in the CG). However, there may be cases in which higher-order beliefs cannot be inferred from the CG. We intend to annotate and model higher-order beliefs in dialog, i.e. beliefs held about other people's belief which are not derivable from CG.

**Error Analysis** Finally, we intend to conduct a more detailed error analysis that could give us more insight to specific issues that need to be addressed in future work.

## Acknowledgments

Markowska's, Rambow's, and Soubki's contributions to this work were supported in part by funding from the Defense Advanced Research Projects Agency (DARPA): Markowska and Rambow under No. HR001120C0037 and PR No. HR0011154158, and Rambow and Soubki under No. HR001122C0034 (CCU). The views, opinions and/or findings expressed are those of the author and should not be interpreted as representing the official views or policies of the Department of Defense or the U.S. Government. Mirroshandel participated in this research while a visitor to the Institute for Advanced Computational Science at Stony Brook University, and gratefully acknowledges its support. We thank: Lee Cohen, Alana Gill, Lawrence Ma, Erica Solis, the annotators for their help in creating this corpus; Ying Ou for creating the interface; Susan Brennan and John Murzaku for useful suggestions and encouragement; and four anonymous reviewers for useful and constructive feedback on the submission.

## Limitations

Since this work introduces a novel language corpus and provides base line system results, we acknowledge several limitations. First, our event formulation procedure does not account for every event that can actually be inferred. For example, if A said *Now we have a king in England, Charles III*, the only event that would be recognized is *Charles III is now the king of England*. Other inferences such as presupposition that Charles' predecessor was not a king will be omitted. Furthermore, we limited anaphora resolutions to personal pronouns only, disregarding other types such as temporal or event anaphora.

One crucial component of our corpus is the representation of updates of CG in the mind of interlocutors. That often resulted in multiple annotations associated with one event, some of which are updates of belief and CG about previous events. Because of the complexity of this procedure, we ignored the updates in our experiments. Finally, we reported only preliminary error analyses for each task that may not provide very detailed insight into the systems performances, and consequently may not be informative enough to clearly understand what needs to be improved.

We have only worked on English; other languages have different strategies for manipulating the CG and we intend to extend this work to more languages.

## Ethics Statement

The annotation procedure developed for the purpose of CG annotation is the first one we are aware of. It reflects on our assumption on how speakers establish and update their common grounds in a dialog, however, we acknowledge that this may not necessarily reflect what actually happens in our mind. As a result, we do not recommend using our procedure to make any important decision about regarding people's (mutual) beliefs. The data we used for the experiments and the inter-annotator agreement evaluations are publicly available. All annotation was done in-house with trained undergraduate students who received credit and/or payment.

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

## A Access to Data and Code

The corpus and code related to experiments are available in the GitHub repository.[1] All implementations use the Python programming language.

## B Belief: different train test splits

The results of the belief classification tests, considering each corpus conversation as the test set. As we mentioned before, four conversations have been used for the tests and these conversations are numbered as "I", "II", "III" and "IV". In the main tests presented in Table7, conversation "IV" is considered as the test set. To check the stability of the results obtained by the models trained on the main test set of the CG body, we also conducted experiments with different train test splits. These experiments done for, FLAN-T5 Fixed Window 2, FLAN-T5 Fixed Window 4, FLAN-T5 Speaker-Based Window. The results of Macro F1 and accuracy of the model considering each conversation

as a test set were calculated and presented in Table 12. To analyze the stability of the results, their average and standard deviation were determined in Table 13.

---

[1] https://github.com/cogstates/2023-emnlp-common-ground

| Models | Bel(A,B) AVG | | | | | |
|---|---|---|---|---|---|---|
| | CT+ F1 | CT- F1 | PS F1 | NB F1 | Macro F1 | Accuracy |
| I is test set | | | | | | |
| FLAN-T5 Fixed Window 2 | 92.00 | 26.50 | 6.25 | 14.25 | 34.75 | 82.00 |
| FLAN-T5 Fixed Window 4 | 93.00 | 33.75 | 7.50 | 11.75 | 36.50 | 84.00 |
| FLAN-T5 Speaker-based window | 90.50 | 27.25 | 8.25 | 18.75 | 36.19 | 80.50 |
| II is test set | | | | | | |
| FLAN-T5 Fixed Window 2 | 87.00 | 15.00 | 32.75 | 10.75 | 36.38 | 73.00 |
| FLAN-T5 Fixed Window 4 | 86.75 | 23.50 | 33.75 | 11.00 | 38.75 | 73.00 |
| FLAN-T5 Speaker-based window | 87.75 | 19.50 | 37.25 | 6.75 | 37.81 | 73.75 |
| III is test set | | | | | | |
| FLAN-T5 Fixed Window 2 | 91.00 | 27.25 | 54.25 | 23.00 | 48.88 | 81.50 |
| FLAN-T5 Fixed Window 4 | 91.00 | 25.00 | 43.50 | 20.50 | 45.00 | 80.00 |
| FLAN-T5 Speaker-based window | 91.50 | 27.00 | 46.25 | 28.75 | 48.38 | 81.25 |
| IV is main test set | | | | | | |
| FLAN-T5 Fixed Window 2 | 86.83 | 34.67 | 28.67 | 17.50 | 41.92 | 74.67 |
| FLAN-T5 Fixed Window 4 | 87.50 | 27.50 | 25.50 | 11.00 | 37.88 | 74.00 |
| FLAN-T5 Speaker-based window | 87.67 | 32.67 | 30.67 | 17.83 | 42.21 | 76.33 |

Table 12: Experimental results of the belief classification with different train test split.

| Models | Bel(A,B) AVG, STD | | | | | |
|---|---|---|---|---|---|---|
| | CT+ F1 | CT- F1 | PS F1 | NB F1 | Macro F1 | Accuracy |
| | AVG, STD | AVG, STD | AVG, STD | AVG, STD | AVG, STD | AVG, STD |
| FLAN-T5 Fixed Window 2 | 89.21, 2.320 | 25.85, 7.033 | 30.48, 17.033 | 16.38, 4.509 | 40.48, 5.528 | 77.79, 4.006 |
| FLAN-T5 Fixed Window 4 | 89.56, 2.552 | 27.44, 3.915 | 27.56, 13.220 | 13.56, 4.017 | 39.53, 3.258 | 77.75, 4.493 |
| FLAN-T5 Speaker-based window | 89.35, 1.684 | 26.60, 4.685 | 30.60, 14.042 | 18.02, 7.790 | 41.15, 4.719 | 77.96, 3.068 |

Table 13: Average and standard deviation of the final results on belief classification task using different train test split.