# OpenReview forum: "Finding Common Ground: Annotating and Predicting Common Ground in Spoken Conversations"
_EMNLP/2023/Conference — EMNLP 2023 Findings_

### Official Review · Reviewer_vhRH · 2023-08-01

**Soundness:** 4

**Excitement:**

3: Ambivalent: It has merits (e.g., it reports state-of-the-art results, the idea is nice), but there are key weaknesses (e.g., it describes incremental work), and it can significantly benefit from another round of revision. However, I won't object to accepting it if my co-reviewers champion it.

**Paper Topic And Main Contributions:**

The paper continues the line of work of factual checking and belief checking in the direction of predicting common grounds. The main two contributions are making available an annotated dataset for the task of CG  precision (they re-annotate an existing dataset). The second contribution is to evaluate the hardness of the CG prediction task by training a classifier and showing its performance.


**Questions For The Authors:**

1. Why can't you view CG, in the way to define it, as fact-checking (or belief checking) where the bank of facts if being updated? If so, why not use a Fact Checking pipeline as a baseline

2. the tasks of predicting humor and satire seem promising avenues to check CG. People understand satire only when they share a broad CG.

3. What doe the numbers in Table 5 represent? It's better to add this to the caption (in all tables) to read tables as stand-alone.

4. For event detection and belief detection, you could have used previous work since these tasks are not new. Why didn't you?


**Reasons To Accept:**

The paper is very well written and easy to follow. The CG dataset is indeed unique and would be used by other researchers to study the CG problem.

**Reasons To Reject:**

The perspective of CG that the authors study is limited (only to beliefs) as they themselves attest. Therefore one can think of their work as an extension of fact-checking where the fact database is being updated with new facts or facts re removed.

**Reproducibility:**

4: Could mostly reproduce the results, but there may be some variation because of sample variance or minor variations in their interpretation of the protocol or method.

**Reviewer Confidence:**

2: Willing to defend my evaluation, but it is fairly likely that I missed some details, didn't understand some central points, or can't be sure about the novelty of the work.

---

> ### Author Rebuttal · Authors · 2023-08-29
>
> We thank the reviewer for their constructive comments, suggestions, and questions.
>
>
> > Q1: Why can't you view CG, in the way to define it, as fact-checking (or belief checking) where the bank of facts is being updated? If so, why not use a Fact Checking pipeline as a baseline
>
> The task of choosing between JA and IN is indeed similar to fact-checking, as long as we have a complete representation of the CG.  However, in our case some shared beliefs come from prior conversations or outside knowledge so there is nothing to check against. Items which are marked as IN need not appear explicitly in the prior text. For the cases where evidence for a belief can be found from the text, a fact-checking inspired approach might work.  In addition, the detection of rejection by the discourse partner (RT) does not appear to have any equivalent in fact-checking.
>
>
> > Q2: the tasks of predicting humor and satire seem promising avenues to check CG. People understand satire only when they share a broad CG.
>
> Yes, those are definitely exciting avenues to extend our work. Thank you for the comment!
>
>
> > Q.3: What do the numbers in Table 5 represent? It's better to add this to the caption (in all tables) to read tables as stand-alone.
>
> As explained in line 351, these numbers denote the length of context employed. In the event generation process, we input the target utterance along with a structured context, as detailed in line 359, into the Flan T5 language model. To illustrate, consider the "FLAN-T5 Fixed Window Size 4" method for event generation, where the context appended to the target utterance encompasses all preceding utterances up to a length of 4. This approach facilitates a comprehensive contextual input for generating events.
> We will take your suggestion into account in any final version of the paper.
>
>
> > Q.4: For event detection and belief detection, you could have used previous work since these tasks are not new. Why didn't you?
>
> As can be seen from Table 1, our event descriptions are not in general substrings of the text of the conversations.  There are different reasons for this (expansion of co-reference, introduction of speech act verbs in certain cases, expansion of short answers – all illustrated in Table 1).  We chose this representation of events to make them both expressive as standalone events, and easy to annotate.  We do not wish to give up our explicit notion of event in order to be able to use previous existing methods, and generative techniques using LLMs are ideally suited for this purpose.  On the other hand, we would need to introduce additional complex machinery to address the other aspects of our event descriptions we just mentioned.
> Even if we did want to identify substrings of the input text, extensive testing on tasks of this nature, including those addressed in current research literature, show that LLM-based generative approaches work better.
>
>
> For belief detection, we will be using existing research in future work, both the corpora and the actual research on predicting belief/factuality.

---

### Official Review · Reviewer_R4T2 · 2023-08-10

**Soundness:** 3

**Excitement:**

3: Ambivalent: It has merits (e.g., it reports state-of-the-art results, the idea is nice), but there are key weaknesses (e.g., it describes incremental work), and it can significantly benefit from another round of revision. However, I won't object to accepting it if my co-reviewers champion it.

**Missing References:**

Work on dialogue systems that deals with modelling BDIs and information state of agents.

**Paper Topic And Main Contributions:**

The paper describes annotation of an existing LDC CALLHOME American Speech corpus with speaker and hearer beliefs and their common ground information states with labels that are described and motivated in the paper. The corpus is validated with ML models based on BERT and FLAN-T5 pre-trained models.

**Questions For The Authors:**

(1) l.121: "Another crucial aspect of the CALLHOME corpus is that the interlocutors know each other very well, which allows us to assume that their CG set is non-empty." Can it be empty at all? There must be something there as soon as interaction begins. Confusing common ground with background knowledge?

Why precisely the CALLHOME corpus? There are several candidates that could be used instead and would full-fill the same criteria given in the explanation or be even better. In l.123 it is stated that it contains 120 dialogues but in Table 4 we see that only 415 + 146 turns have been annotated which come from 4 dialogues (stated somewhere else). Hence, the dialogues are also quite short?

(2) p4, footnote 1: Description of why this is the case?

(3) l.360: If I understand correctly description at this point, the output of the model to detect events is a sequence of generated text? Why not IOB labels?

See also discussion in l.391 and following. These errors would be avoided with the IOB labels.

(4) l.366 taking 4 dialogues for training and 1 dialogue for testing is extremely risky for a discourse task, especially on categories such as beliefs and common ground. While the labels for common ground are objective, the dialogues will be very different in terms of language between individuals and hence the test results might not be representative. You mention that participants were familiar with each other which here presents a challenge since one cannot control for the degree of background knowledge the dyads will be possession prior to establishing the common ground leading to very different language.

Have you tried running the experiment with different dialogue as a test set? I expect this would give very different results.

(5) l.438 Why do you expect that data-augmentation based on translation works best (and not some other method of dialogue augmentation)? It is unintuitive as one would expect that common ground is overtly signalled very differently across different languages.

l.492: Why do you think this method of data augmentation works best?

(6) The section on Future work is quite vague.

Why continue straight away with the audio signal, given that semantic information from text still poses several challenges (described in this paper)? Would that really solve them?

"We also intend to conduct a more detailed error analysis that could give us more insight to specific issues that need to be addressed in future work." What in particular?

**Reasons To Accept:**

The annotation is carefully done and well-motivated and as such the corpus presents a valuable contribution to discourse-annotated corpora that can be used with ML models dealing with text generation (as opposed to dialogue systems).

The approach to dialogue as text opens interesting questions how research on textual models and dialogue systems can learn from each other (but this is not explicitly addressed in the paper).

**Reasons To Reject:**

The authors claim (l.56) that this is the first corpus of common ground which in a very narrow sense is probably true but common ground has been modelled extensively, in different ways, and to different degrees of depth in rule-based and hybrid dialogue systems that the paper does not compare with or refer to (with the exception of work by Traum).

The authors do not seem to be aware of the work on dialogue systems as they state "While there has been much attention paid to the common ground in cognitive science, there has not been much work in natural language processing" which is simply not true as there this a very prominent ACL track.

The main contribution of the paper is therefore the annotation itself.

**Reproducibility:**

3: Could reproduce the results with some difficulty. The settings of parameters are underspecified or subjectively determined; the training/evaluation data are not widely available.

**Reviewer Confidence:**

4: Quite sure. I tried to check the important points carefully. It's unlikely, though conceivable, that I missed something that should affect my ratings.

---

> ### Author Rebuttal · Authors · 2023-08-29
>
> We thank the reviewer for their constructive comments, suggestions, and questions.
>
> > Q(1) “Confusing common ground with background knowledge?”
>
> We acknowledge that one can distinguish between different types of common knowledge in discourse, with background knowledge (BK) including common knowledge based on common cultural, social, or education background. In addition, there is common knowledge shared because of previous conversations, or previous non-verbal experiences, or acquired in the current conversation only.  We have decided to not distinguish among these types of common knowledge (as it can be difficult given our dataset), and refer to all as t​​he common ground (CG).  So in this sense it is true, that it is safe to assume that the CG cannot ever be empty, and we chose our words badly and thank the reviewer for pointing this out.  What we mean this is that there is a distinction between conversation between two strangers and two close friends, such that the CG between the former is likely empty in the sense that it does not contain any non-BK information, and, consequently, hinders and limits freedom of conversation since it does not allow for, for example, proper name resolutions. We will make the distinction between those two concepts clearer. Thank you for pointing that out.
>
> > “Why precisely the CALLHOME corpus? There are several candidates that could be used instead and would full-fill the same criteria
> given in the explanation or be even better. In l.123 it is stated that it contains 120 dialogues but in Table 4 we see that only 415 + 146 turns have been annotated which come from 4 dialogues (stated somewhere else). Hence, the dialogues are also quite short?”
>
>  We specifically chose CALLHOME because it was crucial for us to examine dialogues, as opposed to conversations among multiple interlocutors. Additionally, we wanted the interlocutors to be familiar with each other, which allows for more diversity in linguistic expressions. Finally, it was crucial for us that the conversations were not scripted, and in English. We are not aware of any other corpora that meet all of those requirements.  120 dialogues is the entire CALLHOME corpus, we annotated only 4 dialogues so far, as explained in lines 364.  Furthermore, the dialogues often do not consist of the entire conversation between the interlocutors.
>
> > Q2:  p4, footnote 1: Description of why this is the case?
>
> This footnote is a mistake, thanks for noticing.  We will remove it.
>
>
> > Q.3:  l.360: If I understand correctly, description at this point, the output of the model to detect events is a sequence of generated text? Why not IOB labels?
> See also discussion in l.391 and following. These errors would be avoided with the IOB labels.
>
> As can be seen from Table 1, our event descriptions are not in general substrings of the text of the conversations.  There are different reasons for this (expansion of co-reference, introduction of speech act verbs in certain cases, expansion of short answers – all illustrated in Table 1).  We chose this representation of events to make them both expressive as standalone events, and easy to annotate.  We do not wish to give up our explicit notion of event in order to be able to use IOB, and generative techniques using LLMs are ideally suited for this purpose.  Of course, LLMs may hallucinate, and IOB techniques cannot.  But if we used IOB, we would need to introduce additional complex machinery to address the other aspects of our event descriptions we just mentioned.
> Even if we did want to identify substrings of the input text, extensive testing on tasks of this nature, including those addressed in current research literature, show that LLM-based generative approaches work better.  One interesting idea for future work inspired by this question could be to first use an IOB approach and then an LLM-approach to further expand and transform the extracted passages.
>
>
> > Q.4: Have you tried running the experiment with different dialogue as a test set? I expect this would give very different results.
>
> Actually, it is three training dialogs and one test dialog.  Due to space constraints, in the paper, we purposefully reported the results on a dialogue as a test set with a balanced representation of the minority classes.
> We provide here the results of the two most effective event classification methods (namely, those achieving the highest Macro F1 measure without data augmentation in table 7) in a cross-validation setting, using each of the four conversations as the test dataset in turn. We provide the average F1 and Accuracy scores, and their corresponding standard deviations, across all four folds.
>
>
> Belief Classification with Context using Fixed Window Size 2 method (all numbers are based on %):
>
> Class/Measure:           CT+ F1    |   CT- F1   |   PS F1   |   NB F1   |    Macro F1   |   Accuracy
>
> Average:       89.21      |    25.85    |   30.48    |   16.38    |      77.79       |    40.48
>
> STD:                2.32     |     7.03     |   17.03    |     4.51    |       4.01        |      5.53
>
>
> Belief Classification with Context using Speaker-based Window method:
>
> Class/Measure:           CT+ F1    |   CT- F1   |   PS F1   |   NB F1   |    Macro F1   |   Accuracy
>
> Average:       89.35      |    26.60    |   30.60    |   18.02    |      77.96       |    41.15
>
> STD:                1.68     |     4.69     |   14.04    |     7.79    |       3.07        |      4.72
>
> Our approach yields satisfactory outcomes across these varied train-test partitions.  However, due to the limited occurrences of minority classes (CT-, PS, and NB), the effects of result fluctuations across different train-test splits are pronounced.  We will include these results in any final version.
>
>
> > Q.5: l.438 Why do you expect that data-augmentation based on translation works best (and not some other method of dialogue augmentation)? It is unintuitive as one would expect that common ground is overtly signaled very differently across different languages.
>
> Considering the size of our corpus, we embarked on exploring various data augmentation strategies. Among these, paraphrasing emerged as one approach we adopted. We used GPT-3.5 to generate sentence paraphrases. However, the best outcomes materialized through a distinct avenue: data augmentation via translation; due to the space limit, we have  only reported the results of data augmentation via translation. Notably, instances where we employed translations in French and German, as highlighted in Table 7, exhibited noteworthy performance enhancements.
> This intriguing result might stem from the fact that multilingual language models, like Flan-T5, possess a unique proficiency in comprehending concepts across different languages. Yet, the question at hand necessitates further inquiry, a pursuit we intend to delve into during our forthcoming studies.
>
>
> > Q. 6: The section on Future work is quite vague.
>
> Why continue straight away with the audio signal, given that semantic information from text still poses several challenges (described in this paper)? Would that really solve them?
> "We also intend to conduct a more detailed error analysis that could give us more insight to specific issues that need to be addressed in future work." What in particular?
>  We believe that audio signals will allow for resolution of many semantic and pragmatic ambiguities, especially when it comes to short responses, such as “mhm”, “okay”, “i mean”, “i see”, etc. The error analysis conducted for the purpose of this paper looked at all the errors across all 4 annotated dialogues. It would be interesting to conduct dialogue-internal error analysis to find any potential correlations between the type of vocabulary, syntax, communication style used in a dialogues and types of errors observed.

---

### Official Review · Reviewer_Vosj · 2023-08-11

**Soundness:** 3

**Excitement:**

4: Strong: This paper deepens the understanding of some phenomenon or lowers the barriers to an existing research direction.

**Missing References:**

The difference between the previous works on Theory-of-Mind and common ground in situated dialogues and this work can be discussed under Related work.
1. Theory-of-Mind:

    [a] Michal Kosinski. Theory of mind may have spontaneously emerged in large language models. arXiv preprint arXiv:2302.02083, 2023.364

    [b] Tomer Ullman. Large language models fail on trivial alterations to theory-of-mind tasks. arXiv preprint arXiv:2302.08399, 2023.366

    [c] Damien Sileo and Antoine Lernould. Mindgames: Targeting theory of mind in large language models with dynamic epistemic modal logic. arXiv preprint arXiv:2305.03353, 2023
2. Common ground:

    [a] He He, Anusha Balakrishnan, Mihail Eric, and Percy Liang. Learning symmetric collaborative dialogue agents with dynamic knowledge graph embeddings. Annual Meeting of the Association for Computational Linguistics (ACL), 2017.

    [b] Janosch Haber, Tim Baumgärtner, Ece Takmaz, Lieke Gelderloos, Elia Bruni, and Raquel Fernández. The PhotoBook dataset: Building common ground through visually-grounded dialogue. In Proceedings of the 57th Annual Meeting of the Association for Computational Linguistics, pages 1895–1910, Florence, Italy, July 2019. Association for Computational Linguistics. doi: 10.18653/v1/P19-1184.



**Paper Topic And Main Contributions:**

This paper introduced a new corpus annotated with sentence-level events, beliefs, and common ground regarding the events of the speakers involved in the dialogue. The annotation procedure and inter-annotator agreement are carefully presented. Preliminary experiments are conducted on event extraction, belief classification, and common ground updates type.

**Questions For The Authors:**

1. Ambiguity in mind annotation:
    a. How to distinguish between certainly believe and possibly believe?
2. Experiment set-up clarification:

    a. For experiments in 4.1, what are the results for the speaker-based window?

    b. For experiments in Table 7, is it based on events input or utterance input? For events input, is it the ground truth events or the predicted events?
3. Common ground prediction:

    a. How do context and belief contribute to the common ground prediction? What are the results if sole belief is given as the input?

    c. In Lines 602-605, CG updates depend on the dialogue history. To distinguish between JA and IN, would it be beneficial to add CG history to the input?


**Reasons To Accept:**

1. This paper is the first to approach sentence-level belief prediction and study common ground based on speakers’ beliefs in daily conversation.
2. The corpus annotation involves intensive procedure design and sanity check. The authors clearly presented the definition and designed detailed similarity measures to filter annotators.
3. Comprehensive experiments and analyses have been conducted on each subtask.


**Reasons To Reject:**

1. The corpus covers a detailed annotation of events and interlocutors' beliefs. There are interesting several analyses can be made by taking advantage of these annotations. It can give insights into the model design:

    a. For CG updates, if e is RT, does that mean not e will definitely be JA? Is there a significant dependency between certain annotations?

    b. It also looks like certain words are more correlated to certain belief categories. How do you address the potential bias or its influence on data statistics?

    c. A statistical analysis of the CG and beliefs dependency would be interesting besides the heuristics designed in L527.

    d. Second-order beliefs: as mentioned in L257-266, the common ground depends on not only the first-order belief of the events but also the second-order beliefs of others’ minds. Additional studies can be conducted regarding the relations between common ground and second-order beliefs.


2. I have several clarification questions below.


**Reproducibility:**

3: Could reproduce the results with some difficulty. The settings of parameters are underspecified or subjectively determined; the training/evaluation data are not widely available.

**Reviewer Confidence:**

3: Pretty sure, but there's a chance I missed something. Although I have a good feel for this area in general, I did not carefully check the paper's details, e.g., the math, experimental design, or novelty.

---

> ### Author Rebuttal · Authors · 2023-08-29
>
> We thank the reviewer for the detailed comments and many useful suggestions, and for the thought and care they put into t​​he review.
>
> >> Reasons to reject:
>
> > 1a. “For CG updates, if e is RT, does that mean not e will definitely be JA? Is there a significant dependency between certain annotations?”
>
> The only time where we allow not e to enter the CC after e was annotated RT is in explicit negations of events, e.g. A: “Did he go to Paris last night?” B: “No.” or A: “He went to Paris last night.” B: “No, he didn’t”. In both cases the event of A going to Paris last night will be rejected, and its negation will be added to CG. This solely depends on B’s answer and not on the fact that e was RT, i.e. B’s response “no” is interpreted as “A did not go to Paris last night”. There are cases where we reject an event and do not add its negation, for example, hypotheticals. A: “If I had gone to Paris, I would have visited the Louvre”. While the conditional relation between the events is entered in the CG, both events (that A goes to Paris, and that A will visit the Louvre) are rejected, but their negations will not be added to the speakers CGs.  In general, our approach is to add what is said explicitly, but not what can be inferred (there can be a large number of inferences in general), which however can at times be hard to pin down.  In such cases, we make explicit choices and explain them in the manual.
>
> > 1b. “It also looks like certain words are more correlated to certain belief categories. How do you address the potential bias or its influence on data statistics?”
>
> There definitely exists a set of vocabulary that helps determine the level of belief. For example, the predicate “know” or the adverb “certainly” indicate certain belief in an event. However, our annotation procedure strictly states that while it is allowed to use such semantic cues, we also need to consider other factors, such as syntax, surrounding context, and our world knowledge about humans’ interactions.
>
>
> > 1c. “A statistical analysis of the CG and beliefs dependency would be interesting besides the heuristics designed in L527.”
>
> Yes, this is a good suggestion (thank you!) and we will consider performing such analysis for the final version of the paper.
>
>
> > 1d. “Second-order beliefs: as mentioned in L257-266, the common ground depends on not only the first-order belief of the events but also the second-order beliefs of others’ minds. Additional studies can be conducted regarding the relations between common ground and second-order beliefs.”
>
> The CG annotation allows us to deduce second order beliefs (when there is CG, there is also a second-order belief, and third-order, etc).  Of course there are cases in which we have second order beliefs but NOT common ground (A knows that B does not know that A knows that B knows that A is a murderer, and so on), but we chose for now not to annotate these.  We acknowledge that there can be cases where non-CG second-order beliefs arise, and that we will need to include them, but we have not encountered them in our dialogues yet.
>
> >> Questions for the authors:
>
> > Q1: “Ambiguity in mind annotation: a. How to distinguish between certainly believe and possibly believe?”
>
> There is extensive literature (on belief and factuality) on how to differentiate between these two. It mostly depends on certain lexical items, i.e. predicates in complex clauses that imply uncertainty about the embedded event. For example, “know” vs. “think”. The literature also shows that while distinguishing between different types of such predicates is useful, we also need to consider the context. That means that in our annotations, we take into consideration both the semantic meaning encoded in the main predicates, as well as the annotator’s judgements of the level of belief.
>
>
> > Q2.a:  For experiments in 4.1, what are the results for the speaker-based window?
>
> We conducted a range of diverse experiments; however, due to space constraints, we have focused on presenting the most compelling findings. We did do the experiment you suggest, and its results are provided below. Notably, the results indicate that it does not outperform the fixed window methods with window sizes 2 and 4.
>
> *No Context Event Generation: SBERT = 69.96%
>
> *Speaker Based Window Event Generation: SBERT = 71.53% (not in paper)
>
> *Fixed Window Size 2 Event Generation: SBERT = 72.56%
>
> *Fixed Window Size 4 Event Generation: SBERT = 74%
>
> The Speaker Based approach yields a marginal enhancement when contrasted with the Context-Excluded strategy. However, there was no observable progress compared to the Fixed Window context selection methodologies for this specific task. We would incorporate this result in any final version.
>
> > Q2.b:   For experiments in Table 7, is it based on events input or utterance input? For events input, is it the ground truth events or the predicted events?
>
> As referenced in line 456 of the paper, Table 7 showcases outcomes derived from Events (not utterances) utilizing various contexts as inputs. These events represent ground truth events. We also experimented with utterances as inputs, ultimately concluding that using events yields higher accuracy. For example, the results of Macro F1 of two utterance-based methods are shown in the following, compared to the results of using the events approach:
>
> *Context with Fixed Window size 2, employing the events approach: Macro F1 = 41.92%
>
> *Context with Fixed Window size 2, utilizing the utterances approach: Macro F1 = 34.50%  (not in paper)
>
> *Context with Speaker-based Window, employing the events approach: Macro F1 = 42.21%
>
> *Context with Speaker-based Window, utilizing the utterances approach: Macro F1 = 30.40%  (not in paper)
>
> These results would be integrated into either the appendix or the main text of any camera-ready version of the paper.
>
> Needless to say, if we used predicted events, we would see a reduction in performance.  Furthermore, evaluation is tricky in this case.  We will turn to end-to-end systems and their evaluation in future work.
>
> > Q3.a: How do context and belief contribute to the common ground prediction? What are the results if sole belief is given as the input?
>
> Belief stands as a pivotal determinant in the prediction of Common Ground (CG). Put differently, when interlocutors share identical beliefs, both their CGs tend to be JA or IN; conversely, when beliefs diverge, the CGs tend to be RT.  Neither relation is logically necessary, however.
>
> As the reviewer notes, one of the main challenges in CG prediction is distinguishing between IN and JA categories, whether a given proposition is newly introduced or already exists in the common ground.  The results of Tables 9 (using context and belief), 10 (context only) show that for JA classification, belief is important.  For IN classification, the interpretation of the results is more complex: it seems that if Gold belief is used, more context is better, but if Gold belief is not used, less context is better.  We have no explanation for this pattern.  We note that small changes in results should not be over-interpreted as there are 58 cases of IN in the test data.
>
> To answer the second part of this question, we agree that we should have provided results for a beliefs-only FLAN-T5 experiment, and the results will be documented in any final version of the paper.
>
> > Q3.c:  (No Q3b.)  In Lines 602-605, CG updates depend on the dialogue history. To distinguish between JA and IN, would it be beneficial to add CG history to the input?
>
> The efficiency of distinguishing between "JA" and "IN" through the utilization of CG history is important. In the lines 543 to 554 of the paper, we mention that our heuristic approach introduces a "dialog memory" concept that explicitly incorporates CG history in CG prediction. Conversely, in the T5-based experiments, the inclusion of context in the input implies a more implicit consideration of CG history. Nevertheless, the relatively low performance on IN suggests future experiments where CG history is directly employed as an input feature in the final version of the paper, as you suggest.
>
> >> Missing references:  Thank you for the recommended literature. We will definitely mention it in the related work in the final version of the paper.

---

### Official Review · Reviewer_tfoe · 2023-08-11

**Soundness:** 3

**Excitement:**

3: Ambivalent: It has merits (e.g., it reports state-of-the-art results, the idea is nice), but there are key weaknesses (e.g., it describes incremental work), and it can significantly benefit from another round of revision. However, I won't object to accepting it if my co-reviewers champion it.

**Missing References:**

References on Rational Speech Act (RSA) framework.

**Paper Topic And Main Contributions:**

This paper introduces the Common Ground corpus which annotates the existing LDC CALLHOME American Speech corpus with events, speakers' beliefs, and their shared common beliefs. The paper's main contributions include (1) the release of new data resources, namely, the CG corpus, and (2) initial experimental analyses on the task of common belief classification.

**Questions For The Authors:**

Finding a common ground between two people is not a trivial task. The difficulty can arise from ambiguous syntax structures and/or assumed background knowledge. For example, in the following conversation:
A: "I think the king of France is bald."
B: "That's simply not true!"
Possible events include:
"There is no king of France, so the statement is trivially false."
"There is a king of France, but he's not bald."
In this regard, have the authors thought about the different levels of difficulty in the task?

**Reasons To Accept:**

As large language models have become quite powerful, any new pragmatic task (and dataset) that requires "reading between the lines" is welcomed by the NLP community. The notion of "common ground" is closely related to pragmatics, especially the rational speech act framework. Currently, not many dataset resources exist in this domain, albeit with its growing importance in the field. The CG corpus addresses this missing gap.

**Reasons To Reject:**

The main weakness of the paper is the size of the dataset; it builds upon 4 dialogs (561 utterances) which are very small in size, despite acknowledging the difficulty of annotation. Perhaps the authors could bootstrap from the human-annotated datasets to produce a silver-standard but much larger dataset? Orthogonally, authors could leverage the power of instruction-tuned models and provide a step-by-step guideline (rationale) for the task instead of the current input prompt setting. While the main contribution of the paper is the creation of the dataset, the sheer size of it is too small to be useful, and this is reflected in the experimental results.

A more detailed taxonomy of the CG framework is warranted.

Initial experiments seem limited. An obvious baseline would be modeling via the Rational Speech Act (RSA) framework.

**Reproducibility:**

4: Could mostly reproduce the results, but there may be some variation because of sample variance or minor variations in their interpretation of the protocol or method.

**Reviewer Confidence:**

4: Quite sure. I tried to check the important points carefully. It's unlikely, though conceivable, that I missed something that should affect my ratings.

**Typos Grammar Style And Presentation Improvements:**

Line #208: "me" --> "mean"
Line #360, 361: talks about "X" and "{Y}", but they are not presented in the format.

---

> ### Author Rebuttal · Authors · 2023-08-29
>
> We thank the reviewer for their comments and constructive suggestions.
>
> >> Reasons to reject:
>
> > “The main weakness of the paper is the size of the dataset.”
>
> While the dataset may be small by modern standards it has strong IAA between multiple trained annotators for every utterance. The annotation is of very high quality.
>
> > “the sheer size of it is too small to be useful, and this is reflected in the experimental results”
>
> We do not totally follow how the experimental results demonstrate this. The models trained achieve reasonable performance that may be of use in bootstrapping a larger release as you suggest.  While more data would of course be useful, we also see this as a challenge to machine learning in future work.
>
> > “Initial experiments seem limited. An obvious baseline would be modeling via the Rational Speech Act (RSA) framework.”
>
> Thank you for your valuable suggestion. We will consider incorporating your proposed framework as a further baseline in the final version of the paper. It's worth noting that our decision to utilize Flan-T5 for tackling these challenges is based on thorough testing across comparable tasks, along with a careful review of current research literature. Our findings are consistent with the prevailing consensus that adopting a generative approach with a powerful and advanced language model leads to significantly improved outcomes in contrast to traditional methods.
>
>
>
> >> Questions for the authors:
>
> our approach to annotation should be able to disambiguate such cases. This is because we annotate continuous text, using the cues from the whole dialogue when making annotation decisions. Information gained from surrounding utterances will most likely help the annotator to determine whether “that’s simply not true” refers to the king of France not being bald or France not having a king. Note that our dialogues are among actual people.  If in your example B thought there is no king of France, we sincerely doubt they would simply say “That is not true” but instead (or in addition) perhaps “There is no king of France”.  What is an instructive corner case in philosophical semantics may not be relevant to empirical pragmatics: our corpus consists of everyday conversations. (Also remember that “truth in the world” is not part of this research, we are solely focused on speakers' internal beliefs about events.)

---

### Meta-Review · Area_Chair_AgXy · 2023-09-12

**Recommendation:** 4

**Metareview:**

The main contribution of this paper is to annotate the Callhome corpus with speaker and hearer beliefs and their common ground. The dataset is used for the tasks of event extraction, belief classification  and common ground update classification, and provides baseline performances for each of these tasks.

The reviewers appreciate the dataset as timely to allow for evaluation of a pragmatic task, but note that more comparison to related tasks in dialog systems could be discussed. The reviewers also note the small size of the dataset as the main drawback of this work (4 dialogs, 561 utterances); they also suggest modelling the task using a model that fits the task better, such as a rational speech act model.

Overall, the task is timely and important, but the dataset is small and the modelling not fully convincing, so this seems like a piece of ongoing work that might achieve higher impact if extended.

---

### Decision · Program_Chairs · 2023-10-07

**Decision:**

Accept-Findings

**Comment:**

The main contribution of this paper is to annotate the Callhome corpus with speaker and hearer beliefs and their common ground. The dataset is used for the tasks of event extraction, belief classification  and common ground update classification, and provides baseline performances for each of these tasks.

The reviewers appreciate the dataset as timely to allow for evaluation of a pragmatic task, but note that more comparison to related tasks in dialog systems could be discussed. The reviewers also note the small size of the dataset as the main drawback of this work (4 dialogs, 561 utterances); they also suggest modelling the task using a model that fits the task better, such as a rational speech act model.

Overall, the task is timely and important, but the dataset is small and the modelling not fully convincing, so this seems like a piece of ongoing work that might achieve higher impact if extended.